# Neural Architecture Search by Learning Action Space for Monte Carlo Tree Search

## Abstract

Neural Architecture Search (NAS) has emerged as a promising technique for automatic neural network design. However, existing NAS approaches often utilize manually designed action space, which is not directly related to the performance metric to be optimized (e.g., accuracy). As a result, using manually designed action space to perform NAS often leads to sample-inefficient explorations of architectures and thus can be sub-optimal. In order to improve the sample efficiency, this paper proposes Latent Action Neural Architecture Search (LaNAS), which learns actions to recursively partition the search space into good or bad regions that contain networks with concentrated performance metrics, *i.e.*, low variance. During the search phase, as different architecture search action sequences lead to regions with different performance, the search efficiency can be significantly improved by biasing towards the good regions. On 3 NAS datasets, our experimental results demonstrated that LaNAS is $37\times$, $21.7\times$, $19.3\times$, $10.7\times$, $12.3\times$, $16.5\times$ more sample-efficient than Random Search, Regularized Evolution, Monte Carlo Tree Search, Neural Architecture Optimization, TPE and SMAC, respectively. When applied to the open domain, LaNAS achieves 98.0% accuracy on CIFAR-10 and 75.0% top1 accuracy on ImageNet in only 803 samples, outperforming SOTA AmoebaNet with $33\times$ fewer samples.

## 1 Introduction

During the past two years, there has been a growing interest in Neural Architecture Search (NAS) that aims to automate the laborious process of designing neural networks. Architectures found by NAS have achieved remarkable results in image classification (Zoph and Le (2016); Real et al. (2018)), object detection and segmentation (Ghiasi et al. (2019); Chen et al. (2019); Liu et al. (2019)), as well as other domains such as language tasks (Luong et al. (2018); So et al. (2019)).

Starting from hand-designed discrete model space and action space, NAS utilizes search techniques to explore the search space and find the best performing architectures with respect to a single or multiple objectives (*e.g.*, accuracy, latency, or memory), and preferably with minimal search cost.

However, one common issue faced by the previous works on NAS is that the action space needs to be manually designed. The action space proposed by Zoph et al. (2018) involves sequential actions to construct a network, such as selecting two nodes, and choosing their operations. Other prior works including reinforcement learning based (Zoph and Le (2016); Baker et al. (2016)), evolution-based (Real et al. (2018; 2017)), and MCTS-based (Wang et al. (2019); Negrinho and Gordon (2017)) approaches, all use manually designed action spaces. As suggested in Sciuto et al. (2019); Li and Talwalkar (2019); Xie et al. (2019), action

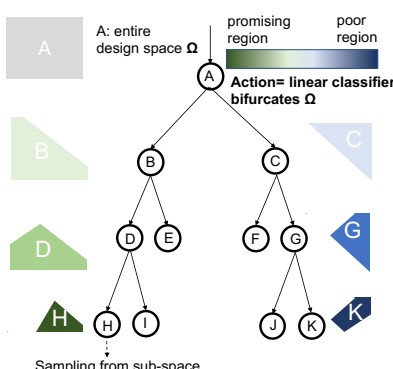

Figure 1: Latent Action Neural Architecture Search: Starting from the entire model space, at each search stage we learn an action (or a set of *linear constraints*) to separate good from bad models for providing distinctive rewards for better searching.

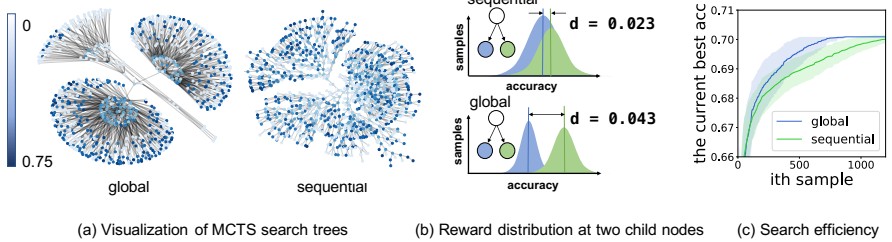

(a) Visualization of MCTS search trees     (b) Reward distribution at two child nodes     (c) Search efficiency

Figure 2: **Illustration of motivation**: (a) visualizes the MCTS search trees using `sequential` and `global` action space. The node value (*i.e.* accuracy) is higher if the color is darker. (b) For a given node, the reward distributions for its children. $d$ is the average distance over all nodes. `global` better separates the search space by network quality, and provides distinctive reward in recognizing a promising path. (c) As a result, `global` finds the best network much faster than `sequential`.

space design alone can be critical to network performance. Furthermore, it is often the case that manually designed action space is not related to the performance that needs to be optimized. In Sec 2, we demonstrate an example where subtly different action space can lead to significantly different search efficiency. Finally, unlike games that generally have a predefined action space (e.g., Atari, Chess and Go), in NAS, it is the final network that matters rather than the specific path of the search, which gives a large playground for action space learning.

Based on the above observations, we propose Latent Action Neural Architecture Search(LaNAS) that learns the action space to maximize search efficiency for performance metrics. Previous methods typically construct an architecture from an empty network by sequentially applying predefined actions, e.g. adding layers, setting the kernel or depth, resulting in a state space that provides undiscriminating rewards to inform the search (Fig. 2a `sequential`). LaNAS takes a dual approach and treats each action as a *linear constraint* that intersects with the search space $\Omega$ to bifurcate it into a good and bad region that contains networks with similar performance metric (Fig. 1). Once multiple actions are recursively applied to the entire search space, the search space is clearly separated (e.g. Fig. 2a `global`, Fig. 4) into regions that provide more informative rewards, thereby greatly improving the search efficiency. To achieve this goal, LaNAS iterates between *learning* and *searching* stage. In the learning stage, each action in LaNAS is learned to partition the model space into high-performance and low-performance regions, to achieve accurate performance prediction. In the searching stage, LaNAS applies MCTS on the learned action space to sample more model architectures. The learned actions provide an informed guide for MCTS to sample from performance-promising regions, while the adaptive exploration in MCTS collects more data to progressively refine the learned actions to zoom into more promising regions for further improving the sample efficiency. The iterative process is jump-started by first collecting a few random samples.

We show that LaNAS yields a tremendous acceleration on a diverse set of benchmark tasks, including publicly available NASBench-101 (420,000 NASNet models trained on CIFAR-10) (Ying et al. (2019)), our self-curated ConvNet-60K (60,000 plain VGG-style ConvNets trained on CIFAR-10), and LSTM-10K (10,000 LSTM cells trained on PTB). Our algorithm consistently finds the best performing architecture on all three tasks with at least an order of fewer samples (on average) than Random Search, Regularized Evolution, MCTS, Neural Architecture Optimization and Bayesian Optimization. In the open domain search scenario, our algorithm finds a network that achieves 98.0% accuracy on CIFAR-10 and 75.0% top1 accuracy (mobile setting) on ImageNet in only 803 samples, using $33\times$ fewer samples and achieving higher accuracy than AmoebaNet (Real et al. (2018)). Moreover, we empirically demonstrate that the learned latent actions can transfer to a new search task to further boost efficiency. Finally, we provide empirical observations to illustrate the search dynamics and analyze the behavior of our approach. We also conduct various ablation studies in together with a partition analysis to provide guidance in determining search hyper-parameters and deploying LaNAS in practice.

## 2   A MOTIVATING EXAMPLE

To demonstrate the importance of action space in NAS, we start with a motivating example. Consider a simple scenario of designing a plain Convolutional Neural Network (CNN) for CIFAR-10 image

Figure 3: **An overview of LaNAS**: LaNAS is an iterative algorithm in which each iteration comprises a search phase and learning phase. The search phase uses MCTS to samples networks, while the learning phase learns a linear model between network hyper-parameters and their accuracies.

classification. The primitive operation is a Conv-ReLU layer. Free structural parameters that can vary include network depth $L = \{1, 2, 3, 4, 5\}$, number of filter channels $C = \{32, 64\}$ and kernel size $K = \{3 \times 3, 5 \times 5\}$. This configuration results in a search space of 1,364 networks. To perform the search, there are two natural choices of the action space: `sequential` and `global`. `sequential` comprises actions in the following order: adding a layer $l$, setting kernel size $K_l$, setting filter channel $C_l$. The actions are repeated $L$ times. On the other hand, `global` uses the following actions instead: {Setting network depth $L$, setting kernel size $K_{1,...,L}$, setting filter channel $C_{1,...,L}$}. For these two action spaces, MCTS is employed to perform the search. Note that both action spaces can cover the entire search space but have very different search trajectories.

Fig. 2(a) visualizes the search for these two action spaces. Actions in `global` clearly separates desired and undesired network clusters, while actions in `sequential` lead to network clusters with a mixture of good or bad networks in terms of performance. As a result, the overall distribution of accuracy along the search path (Fig. 2(b)) shows concentration behavior for `global`, which is not the case for `sequential`. We also demonstrate the overall search performance in Fig.2(c). As shown in the figure, `global` finds desired networks much faster than `sequential`.

This observation suggests that changing the action space can lead to very different search behavior and thus potentially better sample efficiency. In other words, an early exploration on the network depth is critical. Increasing the depth is an optimization direction that can potentially lead to better model accuracy. One might come across a natural question from this motivating example. Is it possible to find a principle way to distinguish a good action space from a bad action space in NAS? Is it possible to *learn an action space* such that it can best fit the performance metric to be optimized?

## 3 LEARNING LATENT ACTIONS

In this section, we describe LaNAS, which comprises two phases: (1) learning phase, and (2) search phase. Fig. 3 presents a high level description of LaNAS, of which the corresponding algorithms are further described in Alg.1.

### 3.1 LEARNING PHASE

In the learning phase at iteration $t$, we have a dataset $D_t = \{(\mathbf{a}_i, v_i)\}$ obtained from previous explorations. Each data point $(\mathbf{a}_i, v_i)$ in $D_t$ has two components: $\mathbf{a}_i$ represents network attributes (e.g., depth, number of filters, kernel size, connectivity, etc) and $v_i$ represents the performance metric estimated from training (or from pre-trained dataset such as NASBench-101). Our goal is to learn a good action space from $D_t$ to guide future exploration as well as to find the model with the desired performance metric efficiently.

Starting from the entire search space $\Omega$, the learnt actions recursively (and greedily) split it into smaller regions such that the estimation of performance metric becomes more accurate. This helps us prune away poor regions as soon as possible and increase the sample efficiency of NAS.

In particular, we model the recursive splitting process as a tree. The root node corresponds to the entire model space $\Omega$, while each tree node $j$ corresponds to a region $\Omega_j$ (Fig. 1). At each tree node $j$, we partition $\Omega_j$ into disjoint regions $\Omega_j = \cup_{k \in (j)} \Omega_k$, such that on each child region $\Omega_k$, the estimation of performance metric $V(\Omega_k)$ is the most accurate (or equivalently, has lowest variance).

At each node $j$, we learn a classifier that embodies an latent action to split the model space $\Omega_j$. The linear classifier takes the portion of the dataset that falls into its own region $D_t \cap \Omega_j$, The performance of a region $\Omega_j$, $V(\Omega_j)$, is estimated by $1/N \sum_{i \in D_t \cap \Omega_j} v_i$. To minimize the variance of $V(\Omega_k)$ for all child nodes, we learn a linear regressor $f_j$ that minimizes $\sum_{i \in D_t \cap \Omega_j} (f_j(\mathbf{a}_i) - v_i)^2$. Once learned, the parameters of $f_j$ and $\hat{V}(\Omega_j)$ form a linear constraint that bifurcates $\Omega_j$ into a good region ($> \hat{V}(\Omega_j)$) and a bad region ($\leq \hat{V}(\Omega_j)$) for sampling. For convenience, the left child always represents the good region. The partition threshold $\hat{V}(\Omega_j)$, combined with $f_j$, forms two latent actions at node $j$, going left ($f_j(\mathbf{a}_i) > \hat{V}(\Omega_j)$) and going right ($f_j(\mathbf{a}_i) \leq \hat{V}(\Omega_j)$).

Note that we need to initialize each node classifier properly with a few random samples to establish initial boundary in the search space. An ablation study on the number of samples for initialization is provided in Fig. 7d.

## 3.2 SEARCH PHASE

Once actions are learned, the search phase follows. The search uses the learned actions to sample more architectures $\mathbf{a}_i$ as well as their performance $v_i$, and store $(\mathbf{a}_i, v_i)$ in dataset $D_t$ to refine the action space in the next learning phase. Note that in the search phase, the tree structure and the parameters of those classifiers are fixed and static. Instead, the search phase learns to decide which region $\Omega_j$ on tree leaves to sample $\mathbf{a}_i$, with a proper balance between the exploration of less known $\Omega_j$ and exploitation of promising $\Omega_j$.

Following the construction of the classifier at each node, a trivial go-left greedy based search strategy can be used to exclusively exploit the most promising $\Omega_k$ defined by the current action space. However, this is not a good strategy since it only *exploits* the current action space, which is learned from the current samples and may not be optimal. There can be good model regions that are hidden in the right (or bad) leaves that need to be explored.

In order to overcome this issue, we use Monte Carlo Tree Search (MCTS) as the search method, which has the characteristics of adaptive exploration, and has shown superior efficiency in various tasks. MCTS keeps track of visiting statistics at each node to achieve the balance between exploitation of existing good region and exploration of new region. In lieu of MCTS, our search phase also has *select*, *sampling* and *backpropagate* stages. LaNAS skips the *expansion* stage in regular MCTS since the connectivity of our search tree is static. When the action space is updated, previous sampled networks and their performance metrics in $D_t$ are reused and redirected to (maybe different) nodes in initializing visitation counts n(s) and node values v(s). When there is no learned action space, we random sample the model space to get jump started.

---

**Algorithm 1:** LaNAS search procedure.

**Data:** specifications of the search domain

1 **Function** $get\_ucb$ ($\bar{X}_{next}$, $n_{next}$, $n_{curt}$ )
2    $c = 0.5$
3    **return** $\frac{\bar{X}_{next}}{n_{next}} + 2c\sqrt{\frac{2log(n_{curt})}{n_{next}}}$ **if** $n_{next} \neq 0$
     **else** $+\infty$
4 **begin**
5    **while** $acc < target$ **do**
6      **for** $n \in Tree.N$ **do**
7        $n.g.train()$
8      **for** $i = 1 \rightarrow \#selects$ **do**
9        $leaf, path = ucb\_select(root)$
10        $constraints = get\_constraints(path)$
11        $network = sampling(constraints)$
12        $acc = network.train()$
13        $back-propagate(network, acc)$

---

**Algorithm 2:** Subroutines in Alg. 1.

1 **Function** $get\_constraints$ (s_path)
2    $constraints = []$
3    **for** $node \in s\_path$ **do**
4      $\mathbf{W}, b = node.g.params(), \bar{X} = node.\bar{X}$
5      **if** $node$ on left **then**
       $constraints.add(\mathbf{W}a + b \geq \bar{X})$ **else**
       $constraints.add(\mathbf{W}a + b < \bar{X})$
6    **return** $constraints$
7 **Function** $ucb\_select$ (c = root)
8    $path = []$
9    **while** $c$ *not leaf* **do**
10      $path.add(c)$
11      $l_{ucb} = get\_ucb(c.left.\bar{X}, c.left.n, c.n)$,
     $r_{ucb} = get\_ucb(c.right.\bar{X}, c.right.n, c.n)$
12      **if** $l_{ucb} > r_{ucb}$ **then** $c = c.left$ **else**
     $c = c.right$
13    **return** $path, c$

---

### 3.3 LaNAS Procedures

**Learning phase**: (Alg.1 line $6 \rightarrow 7$) the learning phase is to train $f_j$ at every node $j$. With more samples, $f$ becomes more accurate, and $V(\Omega_j)$ becomes closer to $\hat{V}(\Omega_j)$. However, as MCTS biases towards selecting good samples, LaNAS will zoom into the promising hyper-space to further improve the sample efficiency. We provided an analysis of such search dynamics in Fig. 4.

**Search phase**: (Alg.1 line $8 \rightarrow 13$) the search phase consists of 3 major steps. 1) ***select w.r.t UCB***: the calculation of UCB follows the standard definition in Auer et al. (2002). We present the equation of $\pi_{UCB}$ in Fig. 3. The input to $\pi_{UCB}$ is the number of visits of current node $n(s)$ and next nodes $n(s, a)$, and the next node value $v(s, a)$, where $s$ stands for a node, and $a$ stands for an action. Therefore, the next node is deterministic given $s$ and $a$. At node j, the number of visits $n(s)$ is determined by the number of samples in $D_t \cap \Omega_j$, while $v(s)$ is determined by $\hat{V}(\Omega_j)$. The selecting policy $\pi_{ucb}$ takes the node with the largest UCB score. Starting from $root$, we follow $\pi_{ucb}$ to traverse down to a leaf (Alg. 3 line 7-13). 2) ***sampling from a leaf***: the *select* traverses a path from the root to a leaf, and each node in the path carries a linear classifier, $\mathbf{W_s} * \mathbf{a}_i + b_s = v(s)$. If the path goes to left child, i.e. $\mathbf{W_s} * \mathbf{a}_i + b_s \geq v(s), a_i \in \Omega$, and $< v(s)$ otherwise. If the original model space is convex, then any selected path corresponds to several linear constraints and thus defines a convex polytope. Sampling in a polytope is known to be *efficient* using Gibbson sampling (Bishop, 2006) or hit-and-run sampler (Smith, 1996). While our implementation uses Gibbson sampling, other method is also available (e.g., hit-and-run[1]). 3) ***back-propagate reward***: after training the sampled network, LaNAS back-propagates the reward, i.e. accuracy, to update the node statistics $n(s)$ and $v(s)$. It also back-propagates the sampled network so that every parent node j keeps the network in $D_t \cap \Omega_j$ for training.

### 3.4 Relationship to other black-box optimizations

**Bayesian Optimization (BO)**. Bayesian Optimization is a popular method for black-box optimization (Snoek et al., 2012; 2015; Wang et al., 2013; Gardner et al., 2014). While BO only uses a fixed kernel which makes it harder to adapt to function with uneven smoothness, LaNAS uses MCTS and online action space learning to dynamically adapt itself towards the important region of the space, and thus achieves better performance in the presence of a decent amount of samples is available.

**Sequential Model-based Global Optimization (SMAC).** SMAC (Hutter et al., 2011) is a generic black box optimization framework when function evaluation is expensive. SMAC (summarized in Appendix A.3) approximates the expensive evaluation with predictions from a surrogate model $\phi$, and proposes the next sample by optimizing an acquisition function (S) on the search space $\Omega$ using $\phi$. LaNAS can be viewed as an instantiation of SMAC framework, but overcomes substantial computational difficulties at different stage. In NAS, the architecture space is huge ($\sim 10^{16}$) and taking $argmax_{\mathbf{a} \in \Omega} S(\mathbf{a}, \phi_t)$ is hard with conventional approaches. Prior works have focused on combining the landscape information of acquisition function with Evolutionary Algorithm (EA) to maximize $S(\mathbf{a}, \phi_t)$ (Bergstra et al., 2011; Kandasamy et al., 2019; Hansen, 2006), as we will show in Fig. 6(b)(c), the high dimensional $\Omega$ in NAS is too large for EA to work well. The contribution of this work is to propose a new search model that utilizes a regression tree as a surrogate model and MCTS as the acquisition function. Instead of solve an optimization every iterations, the tree surrogate learns to partition $f(\mathbf{a_i}), i \in \Omega$ to pinpoint the promising region for the MCTS acquisition function, and it becomes more accurate as the search progresses as shown by Fig. 4), thereby more efficient.

**Regression/Decision Tree.** LaNAS is similar to Decision/Regression tree that recursively splits the data into smaller and smaller bins, and makes decision/prediction at the leaves. While decision/regression tree does it in one shot, LaNAS augments it with online components (e.g., exploration with MCTS, retraining splitting function). As a result, LaNAS is able to drive itself towards interesting regions that initial samples do not touch.

---

[1] https://github.com/fontclos/hitandrun

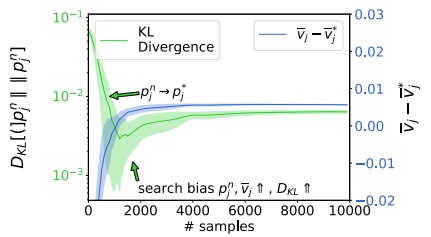 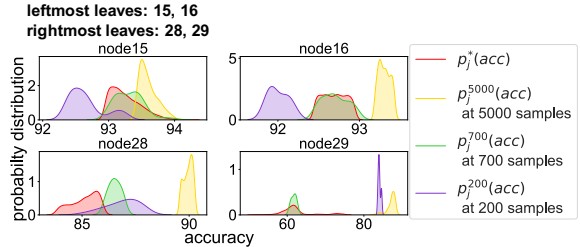

(a) Search dynamics: KL-divergence and mean distance    (b) Search dynamics: sample distribution vs. dataset distribution

Figure 4: **Evaluations of search dynamics**:(a) KL-divergence of $p_j$ and $p_j^*$ dips and bounces back. $\bar{v} - \bar{v}^*$ continues to grow, showing the average metric $\bar{v}$ over different nodes becomes higher when the search progresses. (b) sample distribution $p_j$ approximates dataset distribution $p_j^*$ when the number of samples $n \in [200, 700]$. The search algorithm then zooms into the promising sub-domain, as shown by the growth of $\bar{v}_j$ when $n \in [700, 5000]$.

## 4 EXPERIMENT

We performed extensive experiments on both offline collected benchmark datasets (e.g., NAS-Bench Ying et al. (2019)) and open search domain to validate the effectiveness of LaNAS.

### 4.1 ANALYSIS OF SEARCH ALGORITHM

We analyze LaNAS by looking into the dynamics of sample distributions on tree leaves during the search. We used NASBench-101 in experiments to form a constrained search space that contains 420K models. NASBench-101 provides us with the true distribution of model accuracy, given any subset of model specifications, or equivalently a collection of actions (or constraints). By construction, left nodes contain regions of the good metric while right nodes contain regions of the poor metric. Therefore, at each node $j$, we can construct *reference* distribution $p_j^*(v)$ by training toward the entire dataset to partition the search space into small regions with concentrated performances on leaves. We compare $p_j^*(v)$ with the estimated distribution $p_j^n(v)$, where $n$ is the number of accumulated samples in $D_t \cap \Omega_j$ at the node $j$. Since the *reference* distribution $p_j^*(v)$ is static, comparing $p_j^n(v)$ to $p_j^*(v)$ enables us to see the variations of $\Omega_j$ on tree leaves w.r.t the growing samples. To provide a quantitative view, we also calculated the KL-divergence $D_{KL}[p_j^n \| \| p_j^*]$, and their mean value $\bar{v}_j^* = \mathbb{E}_{p_j^*}[v]$ and $\bar{v}_j = \mathbb{E}_{p_j^n}[v]$.

In our experiments, we use a complete *binary* tree with the height of 5. We label nodes 0-14 as internal nodes, and nodes 15-29 as leaves. By definition, $\bar{v}_{15}^* > \bar{v}_{16}^* ... > \bar{v}_{29}^*$ reflected by $p_{15,16,28,29}$ in Fig. 4b. At the beginning of the search ($n = 200$ for random initialization), $p_{15,16}^{200}$ are expected to be smaller than $p_{15,16}^*$, and $p_{28,29}^{200}$ are expected to be larger than $p_{15,16}^*$; because the tree still learns to partition at that time. With more samples ($n = 700$), $p_j$ starts to approximate $p_j^*$, manifested by the increasing similarity between $p_{15,16,28,29}^{700}$ and $p_{15,16,28,29}^*$, and the decreasing $D_{KL}$ in Fig. 4a. This is because MCTS explores the under-explored regions. As the search continues ($n \to 5000$), LaNAS explores deeper into promising regions and $p_j^n$ is biased toward the region with good performance, deviated from $p_j^*$. As a result, $D_{KL}$ bounces back in Fig. 4a. The mean accuracy of $p_{15}^{700,5000} > p_{16}^{700,5000} > p_{28}^{700,5000} > p_{29}^{700,5000}$ in Fig. 4(b) indicates that LaNAS successfully minimizes the variance of rewards on a search path making architectures with similar metrics concentrated in a region, and LaNAS correctly ranks the regions on tree leaves. These search dynamics show how our model adapts to different stages during the course of the search, and validate its effectiveness for the purpose of searching.

### 4.2 PERFORMANCE ON NAS DATASETS

**Evaluating on NAS Datasets**: We use NASBench-101 (Ying et al. (2019)) as one benchmark that contains over 420K NASNet CNN models. For each network, it records the architectures and their accuracy for fast retrieval by NAS algorithm, avoiding time-consuming model retraining. In addition, we construct two more datasets for benchmarking, ConvNet-60K (60K plain ConvNet models, VGG-style, no residual connections, trained on CIFAR-10) and LSTM-10K (10K LSTM cells trained on

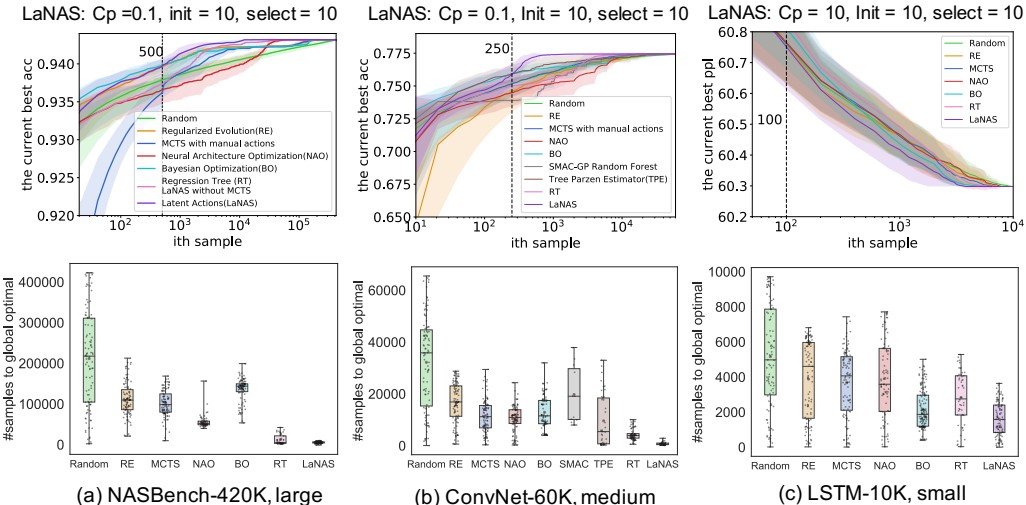

Figure 5: **Evaluations of sample-efficiency** on NASBench, ConvNet-60K and LSTM-10K. Each search algorithm is repeated 100 times on each datasets with different random seeds. The top row shows the time-course of the search algorithms (current best accuracy with interquartile range), while the bottom row illustrates the number of samples to reach the global optimum. NASBench-420K limits the edges $\leq 9$ and LSTM-10K limits the edges = 9, rendering a huge number of architectures untrained in the search space. We compared to SMAC and TPE on ConvNet-60k as ConvNet-60k contains the accuracy of all the architectures in the search space, .

PTB) to further validate the effectiveness of LaNAS. In each task, we perform an independent search for 100 times with different random seeds. The mean performance, along with the 25th and 75th percentile, is shown in Fig.5.

**Baselines**: we compare LaNAS with a diverse types of search algorithms, from Random Search, gradient based NAS, evolutionary algorithms, to Monte Carlo Tree Search (MCTS), and with a strong focus on Bayesian Optimization. Random Search finds the global optimum in expected $n/2$ samples. Regularized Evolution (Real et al. (2019)) is a type of evolutionary algorithm that achieves SoTA performance for image recognition. Our implementation of Regularized Evolution is from NASBench-101[2]. We used AlphaX [3] as our MCTS implementation. It adopts a manual action space that incrementally builds up a network, and the predictions is set to 1 per rollout. Neural Architecture Optimization(NAO) (Luo et al. (2018)) is a gradient based NAS method that finds the architecture on a performance predictor in a continuous embedding space with gradient descent. We adopted NAO from its public release[4]. Regression Tree removes MCTS from LaNAS, and RT always sample from the most promising node. Bayesian Optimization(BO) is a competitive method for the black box optimization. We implemented BO that used Gaussian Process as the surrogate model and Expected Improvement (EI) as the acquisition function based on here[5]. However, the vanilla BO suffers from the scaling issue that it incurs $O(n^3)$ cost, where n is the number of samples. To mitigate this issue, we apply BO on collected samples using a similar strategy to SGD that splits the entire samples into batches, then iteratively fits BO at a batch size of 2000 for 2 epochs on NASBench-420K and ConvNet-60K. We also compared LaNAS to other BO variants that does not have the cubic scaling issue such as TPE (Bergstra et al. (2011)) from HyperOpt [6], and SMAC-random forest (Hutter et al. (2013)) from SMAC [7]. Additionally, we compared to the latest BO package Dragonfly (Kandasamy et al. (2019))[8]. Since Dragonfly is too expensive to collect $10^4$ samples, we only compared to Dragonfly in Fig 6(a)(b).

---

[2]https://github.com/google-research/nasbench/blob/master/NASBench.ipynb

[3]https://github.com/linnanwang/AlphaX-NASBench101

[4]https://github.com/renqianluo/NAO

[5]http://krasserm.github.io/2018/03/21/bayesian-optimization/

[6]https://github.com/hyperopt/hyperopt

[7]https://github.com/automl/SMAC3

[8]https://github.com/dragonfly/dragonfly

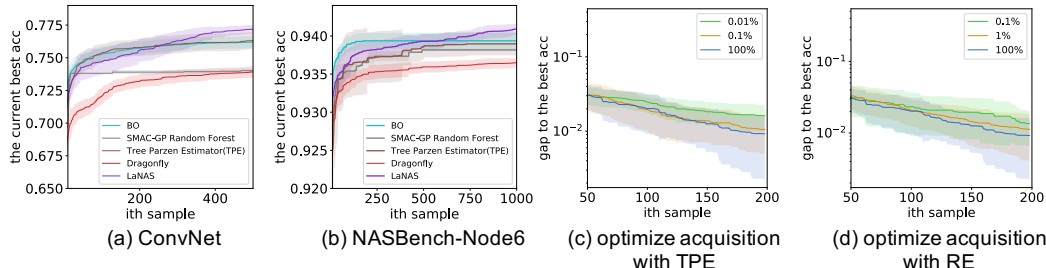

Figure 6: **(a-b) Comparing LaNAS to several mainstream BO methods**: though BO and TPE works well when samples $\in [0, 200]$ on ConvNet and when samples $\in [0, 500]$ on NASBench-Node6 (a subset of NASBench containing 62010 6-nodes networks), LaNAS quickly dominates the performance afterwards. This highlights the different focuses of LaNAS and BO methods. **(c-d)illustrations of the limitation of BO methods**: optimizing the acquisition function with Tree Parzen Estimator (TPE) and Regularized Evolution (RE) using different budgets, that represent the max number of queries to a surrogate model in optimizing the acquisition. We denote the budgets as the percentages of ConvNet dataset.

**Analysis of Results:** Fig. 5 demonstrates that LaNAS consistently outperforms the baselines by significant margins on three separate tasks. Particularly, on NASBench, LaNAS is on average using **37x**, **21.7x**, **19.3x**, **10.7x**, **25.6x** fewer samples than `Random Search`, `Regularized Evolution`, `MCTS`, `NAO` and `Bayesian Optimization` to find the global optimum. On ConvNet-60K, LaNAS used **12.3x**, **16.5x** and **25.3x** fewer samples than `TPE`, `SMAC-random forest` and our relaxed BO. On LSTM, LaNAS still performs the best despite that the dataset is small. Additionally, LaNAS is **2.3x**, **5.4x**, **1.8x** faster than `Regression Tree` on NASBench, ConvNet and LSTM, demonstrating the effectiveness of MCTS to LaNAS. The substantial speedup w.r.t `MCTS` that uses a fixed action space also validates the effectiveness of learned actions.

To highlight the different focuses of BO methods and LaNAS, we also compared the first 500 samples on ConvNet-60K and the first 1000 samples on NASBench-Node6. NASBench-Node6 is a subset of NASBench that contains networks with nodes=6. Since edges in NASBench $\leq 9$, NASBench-Node6 have most networks trained in the search space. Fig. 6 indicates and TPE and BO (no relaxation) are fast in the first few hundreds samples, but LaNAS quickly catches up afterwards. This indicates BO and TPE give good results under limited budget, while LaNAS intends to find the global optimum in the least samples.

In LaNAS, the action space is learned so that in each node, the representing search space can be partitioned into a separate good/bad region using a linear classifier based on the network attributes (e.g., depth/width of the network). With the learned action space, the resulting search space clearly separates good/bad regions, and the search is biased towards the regions that contain better architectures, therefore becoming more efficient. Besides, with more samples, Fig. 4 shows LaNAS zooms in the promising regions pinpointing more promising regions to further improve the efficiency. `Random Search` gives poor performance, in particular in a large search space like NASBench. `Regularized Evolution` utilizes a static exploration strategy that maintains a pool of top 500 architectures for random mutations, which is not guided by previous search experience. BO is worse than LaNAS on NASBench and ConvNet due to our training relaxation. Though `Bayesian Optimization(BO)` (no relaxation) shows slightly worse performance than LaNAS on small LSTM-10K, it maximizes the acquisition on the entire search space which is not possible for high dimensional NAS problem that a typical search space contains over $10^{17}$ samples. To show the influence from the quality of acquisition maximization, we also upgraded the vanilla BO to maximize the acquisition function with Regularized Evolution and TPE. The baseline is the case that using the surrogate to predict all (100%) the architectures in the search space, as in the vanilla BO on LSTM. Fig. 6(c-d) shows the apparent performance deterioration after reducing the optimization budget. Please noted evaluating 0.01% or $10^{13}$ architectures in the search space is still not practical for NAS.

## 4.3 PERFORMANCE ON OPEN DOMAIN SEARCH

To be consistent with existing works, we also evaluated LaNAS on the NASNet open search domain.

**Search space**: Our search space is consistent with the widely adopted NASNet (Zoph et al. (2018)). The operations are 3x3, 5x5 and 7x7 max pool, 3x3, 5x5, 7x7 depth-separable conv, 1x1 convolution

Table 1: Results on CIFAR-10, c/o is cutout.

| Model | Filters | Params | Top1 err | M | GPU days |
|---|---|---|---|---|---|
| NASNet-A+c/o | 32 | 3.3 M | 2.65 | 20000 | 2000 |
| AmoebaNet-B+c/o | 36 | 2.8 M | $2.55_{\pm 0.05}$ | 27000 | 3150 |
| PNASNet-5 | 48 | 3.2 M | $3.41_{\pm 0.09}$ | 1160 | 225 |
| NAO | 36 | 10.6 M | 3.18 | 1000 | 200 |
| ENAS+c/o | 36 | 4.6 M | 2.89 | - | 0.45 |
| DARTS+c/o | 36 | 3.3 M | $2.76_{\pm 0.09}$ | - | 1.5 |
| BayesNAS+c/o | 32 | 3.4 M | $2.81_{\pm 0.04}$ | - | 0.2 |
| ASNG-NAS+c/o | 32 | 3.9 M | $2.83_{\pm 0.14}$ | - | 0.11 |
| LaNet | 32 | 3.2 M | $\mathbf{3.13}_{\pm 0.03}$ | 803 | 150 |
| LaNet+c/o | 32 | 3.2 M | $\mathbf{2.53}_{\pm 0.05}$ | 803 | 150 |
| NAO+c/o | 128 | 128.0 M | 2.11 | 1000 | 200 |
| AmoebaNet-B+c/o | 128 | 34.9 M | $2.13_{\pm 0.04}$ | 27000 | 3150 |
| LaNet+c/o | 128 | 38.7 M | $\mathbf{1.99}_{\pm 0.02}$ | 803 | 150 |

M: number of samples selected.

Table 2: Results on ImageNet (mobile setting)

| Model | FLOPs | Params | top1 / top5 err |
|---|---|---|---|
| NASNet-A (Zoph et al. (2018)) | 564M | 5.3 M | 26.0 / 8.4 |
| NASNet-B (Zoph et al. (2018)) | 488M | 5.3 M | 27.2 / 8.7 |
| NASNet-C (Zoph et al. (2018)) | 558M | 4.9 M | 27.5 / 9.0 |
| AmoebaNet-A (Real et al. (2018)) | 555M | 5.1 M | 25.5 / 8.0 |
| AmoebaNet-B (Real et al. (2018)) | 555M | 5.3 M | 26.0 / 8.5 |
| AmoebaNet-C (Real et al. (2018)) | 570M | 6.4 M | **24.3 / 7.6** |
| PNASNet-5 (Liu et al. (2018a)) | 588M | 5.1 M | 25.8 / 8.1 |
| DARTS (Liu et al. (2018b)) | 574M | 4.7 M | 26.7 / 8.7 |
| FBNet-C (Wu et al. (2018)) | 375M | 5.5 M | 25.1 / - |
| RandWire-WS (Xie et al. (2019)) | 583M | 5.6 M | 25.3 / 7.8 |
| BayesNAS (Zhou et al. (2019)) | - | 3.9 M | 26.5 / 8.9 |
| LaNet | 570M | 5.1 M | **25.0 / 7.7** |

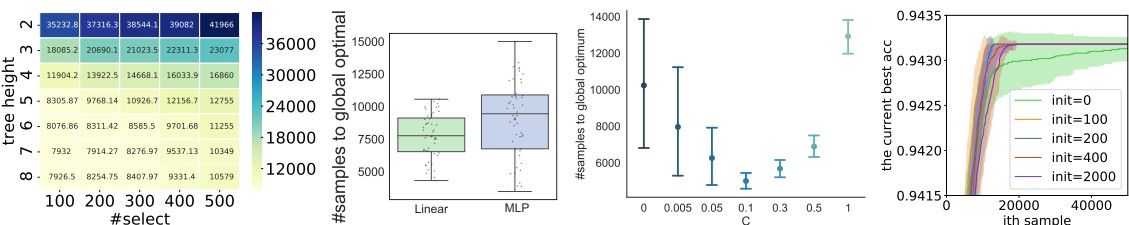

(a) tree height vs. #select, $c$=0.5    (b) choice of classifier    (c) performance at different C    (d) #samples for init

Figure 7: **Ablation study**: (a) the effect of different tree heights and #select in MCTS. Number in each entry is #samples to reach global optimal. (b) the choice of predictor for splitting search space. (c) the effect of C in UCB toward the performance on nasbench. (d) the effect of #samples for initialization toward the search performance.

and identity. The number of nodes within a cell is 7, and we constructed the network by stacking a cell 6 times. Existing works in NAS tends to have very different sizes of search space according to Table.3 in Radosavovic et al. (2019). Though we follow NASBench-101 and PNAS to share the normal and reduction cell in constructing the network, our search space contains $4.36 \times 10^{17}$ architectures, which is decent large for evaluations.

**Experiment setup**: During the search phase, we set filters to 32 and pre-trained an architecture for 120 epochs. Then we selected top-20 architectures collected from the search, and re-trained them for 600 epochs to acquire the final accuracy in Table. 1. We reused the training logic from `DARTS` and their training settings. The height of search tree is 8; we used 200 random samples for the initialization, and #select = 50.

Table. 1 compares our results in the context of searching NASNet style architecture on CIFAR-10. We found the best performing architecture (LaNet) at the 803th sample. LaNet demonstrates an average accuracy of 97.47% (#filters = 32, #params = 3.22M) and 98.01% (#filters = 128, #params = 38.7M), which is better than all existing NAS-based results on CIFAR-10. It is worth noting that we achieved this accuracy with 33x fewer samples than AmoebaNet. Though one-shot learning methods, e.g. Liu et al. (2018b); Zhou et al. (2019); Akimoto et al. (2019), and their weight sharing variants, e.g. Pham et al. (2018), exhibit weaker performance, they render much lower search costs for greatly reducing network evaluation costs with techniques such as transfer learning or approximating final weights with only 1 training iteration. Fig. 5 indicates the search model is cheap to run, but evaluating networks during the search costs the majority of GPU days. Both `NAO` and `ENAS` show that weight sharing has greatly sped up network evaluations; our approach can also benefit from weight sharing to reduce the end-to-end search time. On the other hand, one shot learning methods show a small difference to LaNAS in accuracy, e.g. $-0.8\%$, at much less cost. However, as suggested in Fig. 5, almost 99.9% search costs, i.e. samples, is at improving the last 1% accuracy. Therefore, one-shot learning is fast to yield a reasonable result, while LaNAS aims at giving the best accuracy with the fewest samples.

## 4.4 TRANSFER LEARNING

**Transfer LaNet to ImageNet**: Transferring the best performing architecture (found through searching) from CIFAR10 to ImageNet has already been a standard technique. Following the mobile setting (Zoph et al. (2018)), Table. 2 shows that LaNet found on CIFAR-10, when transferred to ImageNet mobile setting (FLOPs are constrained under 600M), achieves competitive performance.

**Intra-task latent action transfer**: We learn actions from a subset (1%, 5%, 10%, 20% 60%) of NASBench and test their transferability to the

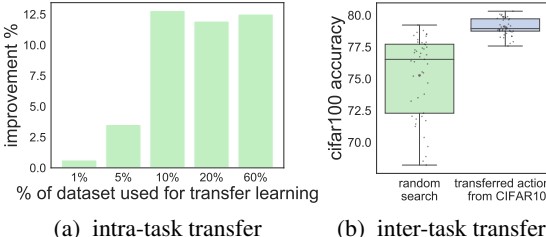

(a) intra-task transfer       (b) inter-task transfer

Figure 8: **Latent actions transfer**: learned latent actions can generalize within the same task or across different tasks, to further boost search efficiency.

remaining dataset, as shown in Fig. 8a. Interestingly, the improvement remain steady after 10%. Consistent with Fig. 4, it is enough to use 10% of the samples to learn the action space.

**Inter-task latent action transfer**: We compare 100 architectures selected by LaNAS from sec.4.3 (on CIFAR10) with 100 random trials. Networks are trained for 100 epochs on CIFAR-100 and their performances are compared. Fig. 8b indicates that inter-task action transfer is also beneficial.

## 4.5 ABLATION STUDIES

**The effect of tree height and #selects**: Fig. 7a relates tree height ($h$) and the number of selects (#selects) to the search performance. In Fig. 7a, each entry represents #samples to achieve optimality on NASBench, averaged over 100 runs. A deeper tree leads to better performance, since the model space is partitioned by more leaves. Similarly, small #select results in more frequent updates of action space allowing the tree to make up-to-date decisions, and thus leads to improvement. On the other hand, the number of classifiers increases exponentially as the tree goes deeper, and a small #selects incurs frequent learning phase. Therefore, both can significantly increase the computation cost.

**Choice of classifiers**: Fig.7b shows that using a linear classifier performs better than an multi-layer perceptron (MLP) classifier. This indicates that adding complexity to decision boundary of actions may not help with the performance. Conversely, performance can get degraded due to potentially higher difficulties in optimization.

**#samples for initialization**:We need to initialize each node classifier properly with a few samples to establish the initial boundary in the search space. As shown in Fig.7d, cold start is necessary (init > 0 is better than init = 0). Also, small init=100-400 converges to top 5% performance much faster than init=2000, while init=2000 gets the best performance faster (Fig. 7d).

**The effect of $c$ in UCB**: Fig. 7c shows that the exploration term, $2c\sqrt{\frac{2log(n_{curt})}{n_{next}}}$, improves the overall performance as $c$ increases from 0 to 0.1, while using a large $c$, e.g. 0.5 and 1, is not desired for over-exploring. Each values of $c$ represents the LaNAS performance on NASBench in 100 runs. Please noted LaNAS is still the leading algorithm on NASBench even using a sub-optimum $c$ (e.g. $c$ = 0 or 1). We find that setting $c$ to $0.1 * max(\bar{X})$ empirically works well on 3 datasets in Fig.5.

## 5 FUTURE WORK

Recent work on shared model (Luo et al. (2018); Pham et al. (2018)) improves the training efficiency by reusing trained components from the similar previously explored architectures, e.g., weight sharing (Liu et al. (2018b); Luo et al. (2018); Pham et al. (2018)). Our work focuses on sample-efficiency and is complementary to the above techniques.

To encourage reproducibility in NAS research, various of architecture search baselines have been discussed in Sciuto et al. (2019); Li and Talwalkar (2019). We will also open source the proposed framework, together with the three NAS benchmark datasets used in our experiments.

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

# A APPENDIX

## A.1 NAS DATASETS AND EXPERIMENT SETUP:

NAS dataset enables directly querying a model's performance, e.g. accuracy. This allows for truly evaluating a search algorithm by repeating hundreds of independent searches without involving the actual training. NASBench-101 Ying et al. (2019) is the only publicly available NAS dataset that contains over 420K DAG style networks for image recognition. However, a search algorithm might overfit NASBench-101, losing the generality. This motivates us to collect another 2 NAS datasets, one is for image recognition using sequential CNN and the other is for language modeling using LSTM.

**Collecting ConvNet-60K dataset**: following a similar set in collecting 1,364 networks in sec.2, free structural parameters that can vary are: network depth $D = \{1, 2, 3, 4, 5, 6, 7, 8\}$, number of filter channels $C = \{32, 64, 96\}$ and kernel size $K = \{3 \times 3, 5 \times 5\}$. We train every possible architecture 100 epochs, and collect their final test accuracy in the dataset.

**Collecting LSTM-10K dataset**: following a similar LSTM cell definition in Pham et al. (2018), we represent a LSTM cell with a connectivity matrix and a node list of operators. The choice of operators is limited to either a fully connected layer followed by RELU or an identity layer, while the connectivity is limited to 6 nodes. We randomly sampled 10K architectures from this constrained search domain, and training them following the exact same setup in Liu et al. (2018b). Then we keep the test perplexity as the performance metric for NAS.

## A.2 OPEN DOMAIN SEARCH AND EXPERIMENT SETUP:

The best convolutional cell found by LaNAS is visualized below. We constructed LaNet by stacking this cell 6 times. *arg min

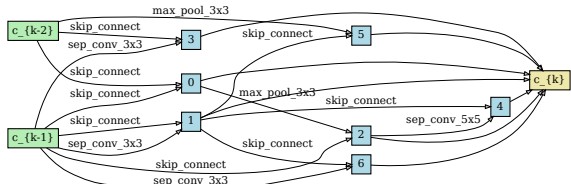

## A.3 SEQUENTIAL MODEL-BASED GLOBAL OPTIMIZATION (SMBO):

---
**Algorithm 3:** psudo-codes for Sequential Model-based Global Optimization (SMBO).

---
1 **INPUT:** $D_t$ - set of collected samples, $\Omega$ - search space, $S$ - acquisition function, $\phi$ - surrogate model
2 **Function** $SMBO (f, M_0, T, S)$
3      $D_t \leftarrow \emptyset$
4      **for** $t \leftarrow 1 : T$ **do**
5          $\mathbf{a}^* \leftarrow argmax_{\mathbf{a} \in \Omega} S(\mathbf{a}, \phi_t)$   ◁ *hard to evaluate in NAS due to large $\Omega$*
6          *evaluate* $f(\mathbf{a}^*)$          ◁ *expensive to general applications*
7          $D_{t+1} = D_t \cup (\mathbf{a}^*, f(\mathbf{a}^*))$
8          $\phi_{t+1} = $ *fit* $\phi_t$ *on* $D_{t+1}$
9      **return** $D_t$

---

## A.4 PARTITION ANALYSIS

The sample efficiency, i.e. the number of samples to find the global optimal, is closely related to the partition quality of each tree nodes. Here we seek an upper bound for the number of samples in the leftmost leaf (the most promising region) to characterize the sample efficiency.

**Assumption 1** *Given a search domain $\Omega$ containing finite samples $N$, there exists a probabilistic density $f$ such that $P(a < v < b) = \int_a^b f(v)dv$, where $v$ is the performance of a network* **a***.*

With this assumption, we can count the number of networks in the accuracy range of $[a, b]$ by $N * P(a \leq v \leq b)$. Since $v \in [0, 1]$ and $\sigma_(v) < \infty$, the following holds (Mallows (1991))

$$|E(\overline{v} - M_v)| < \sigma_v \tag{1}$$

$\overline{v}$ is the mean performance in $\Omega$, and $M_v$ is the median performance. Note $v \in [0, 1]$, and let's denote $\epsilon = |\hat{v} - \overline{v}|$. Therefore, the maximal distance from $\hat{v}$ to $M_v$ is $\epsilon + \sigma_v$; and the number of networks falling between $\hat{v}$ and $M_v$ is $N * max(\int_{\hat{v}-\epsilon-\sigma_v}^{M_v} f(v)dv, \int_{M_v}^{\hat{v}+\epsilon+\sigma_v} f(v)dv)$, denoted as $\delta$. Therefore, the root partitions $\Omega$ into two sets that have $\leq \frac{N}{2} + \delta$ architectures.

**Theorem 1** *Given a search tree of height = $h$, the sub-domain represented by the leftmost leaf contains at most $2 * \delta_{max}(1 - \frac{1}{2^h}) + \frac{N}{2^h}$ architectures, and $\delta_{max}$ is the largest partition error from the node on the leftmost path.*

The theorem indicates that LaNAS is approximating the global optimum at the speed of $N/2^h$, suggesting 1) the performance improvement will remain near plateau as $h \uparrow$ (verified by Fig 7a), while the computational costs ($2^h - 1$ nodes) exponentially increase; 2) the performance improvement w.r.t random search (cost $\sim N/2$) is more obvious on a large search space (verified by Fig.5 (a)→(c)).

*Proof of Theorem*: In the worst scenario, the left child is always assigned with the large partition; and let's recursively apply this all the way down to the leftmost leaf $h$ times, resulting in $\delta^h + \frac{\delta^{h-1}}{2} + \frac{\delta^{h-2}}{2^2} + ... + \frac{N}{2^h} \leq 2 * \delta_{max}(1 - \frac{1}{2^h}) + \frac{N}{2^h}$. $\delta$ is related to $\epsilon$ and $\sigma_v$; note $\delta \downarrow$ with more samples as $\epsilon \downarrow$, and $\sigma_v$ becomes more accurate.

## A.5    ORIGINAL FIG.5 IN LOG SCALE

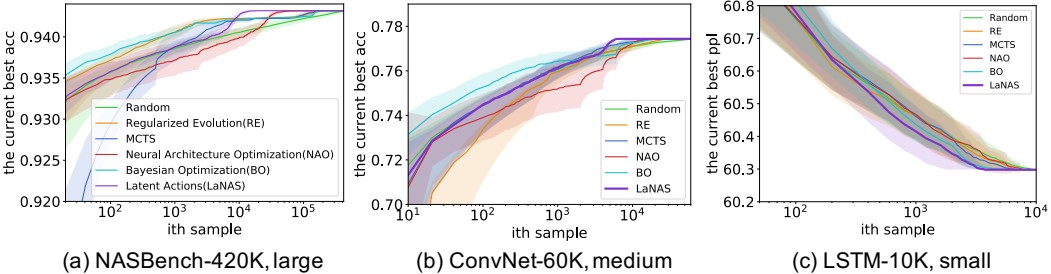

(a) NASBench-420K, large       (b) ConvNet-60K, medium       (c) LSTM-10K, small

Figure 9: previous Fig.5 in log scale.

