# OpenReview forum: "Neural Architecture Search by Learning Action Space for Monte Carlo Tree Search"
_ICLR.cc/2020/Conference — Reject_

### Official Review · AnonReviewer1 · 2019-10-23
**Official Blind Review #1**

**Rating:** 3

**Review:**

In this paper, the authors propose a strategy for neural architecture search. The basic idea is effectively to model the accuracy of architectures in the search space, and use this model to select subsequent architectures with a MCTS-like procedure.

Overall, my primary concern with the paper is a lack of context in the larger field of model based optimization. To be clear, the authors' method is clearly an instantiation of model based optimization. However, much of the paper is arguably written as though this needs to be invented from first principles. For example, much of section 3 is arguably a specific instantiation of the basic model based optimization loop, and much of the discussion on global versus sequential search exists in this literature as well.

I believe the paper would be greatly improved by (1) providing this context, and (2) explaining the authors' approach within this context. Much of the discussion contrasting arbitrary action spaces with handcrafted ones are somewhat lost in the actual experimental setup: For example, the ConvNet-60K and LSTM-10K datasets have well specified parameter spaces. Beyond this, I'd like the authors to contrast the surrogate tree model used with simple CART trees: the fitting procedure in section 3.1 is quite similar to standard methods used to fit regression trees.

Additionally in the same context, a significant amount of related work is missing. The use of tree models for model based optimization have been considered before (e.g., SMAC), although the MCTS acquisition with a single tree surrogate is novel as far as I am aware. Other recent methods in model based optimization exist, including those with specific application to architecture search that explicitly outperform the basic Bayesian optimization algorithm (e.g. NASBot), and I'm therefore not sure if the comparison to the most basic instantiation of BayesOpt is appropriate.

Beyond this, the experimental performance of the authors' method seems quite good on the tasks considered, and at least a substantial subset of the baselines considered are recent. I would therefore not be upset to see the paper published and would be willing to increase my score; however, I believe the framing of the paper needs substantial imrpovement.

**Experience Assessment:**

I have read many papers in this area.

**Review Assessment: Checking Correctness Of Derivations And Theory:**

I assessed the sensibility of the derivations and theory.

**Review Assessment: Checking Correctness Of Experiments:**

I assessed the sensibility of the experiments.

**Review Assessment: Thoroughness In Paper Reading:**

I read the paper at least twice and used my best judgement in assessing the paper.

---

> ### Author Response · Authors · 2019-11-13
> **thank you, here is our answers**
>
> ==>1. lack of context in the larger field of model based optimization. Explaining the authors' approach within this context.
>
> We are very grateful to this valuable feedback; we completely agree that putting our LaNAS framework into the larger context of model based optimization could provide better intuitions, more historical contexts, and an overall better position for this paper. We will be happy to adopt this perspective to frame our paper, develop the methods section and add more discussions to previous work (we added a preliminary Sec 3.4).
> We hope that with this largely overlooked perspective, we can push forward both the NAS and model based optimization research by linking two communities together, extending the application scopes and inspiring new breakthroughs.
>
> ==>2. Much of the discussion contrasting arbitrary action spaces with handcrafted ones are somewhat lost in the actual experimental setup:
>
> Actually, we did compare MCTS with fixed actions (e.g. adding layers, tuning kernels) to MCTS with learned actions (LaNAS) in Fig.5, and LaNAS is at least an order of faster than fixed-action based MCTS on 3 datasets.
>
> In the updated version, we have clarified that MCTS used a fixed action space.
>
> ==>3. Beyond this, I'd like the authors to contrast the surrogate tree model used with simple CART trees.
>
> Excellent suggestions! We have provided Regression Tree that removes MCTS from LaNAS in Fig.5. The results show MCTS is indeed a critical component to the search performance.
>
> ==>4. The use of tree models for model based optimization have been considered before (e.g., SMAC), although the MCTS acquisition with a single tree surrogate is novel as far as I am aware.
>
> Random Forest (RF) in SMAC is to estimate the mean and variance for the predictive distribution to avoid the cubic scaling problem [1], while the regression tree in LaNAS is to partition the search space.
>
> [1] Bayesian Optimization Primer
> https://app.sigopt.com/static/pdf/SigOpt_Bayesian_Optimization_Primer.pdf
>
> ==>5. Other recent methods in model based optimization exist, including those with specific application to architecture search that explicitly outperform the basic Bayesian optimization algorithm (e.g. NASBot), and I'm therefore not sure if the comparison to the most basic instantiation of BayesOpt is appropriate.
>
> Thank you for your suggestions, the experimental section in our updated paper adds several mainstream BO variants including SMAC-random forest, TPE from HyperOpt and the latest Dragonfly to cover a wide spectrum of bayes methods. We didn’t compare against NASBot but compare against Dragonfly, which claims better results than NASBot (Fig.11 in [1]). LaNAS still consistently outperforms SMAC, TPE and Dragonfly as shown by Fig.5(b) and Fig.6(a)(b). We skip Dragonfly on Fig.5 as it is too slow to collect 10^4 samples.
>
> Our answers to “the key limitations of bayesian optimization to NAS” in the thread of general answer to all reviewers also provide a detailed explanation to the key limitation of BO methods when applied to NAS.
>
> [1] Kandasamy, Kirthevasan, et al. "Tuning Hyperparameters without Grad Students: Scalable and Robust Bayesian Optimisation with Dragonfly." arXiv preprint arXiv:1903.06694 (2019).

---

### Official Review · AnonReviewer2 · 2019-10-30
**Official Blind Review #2**

**Rating:** 3

**Review:**

This paper introduces a Neural Architecture Search algorithm that attempts to solve the problems of the existing NAS only utilizing manually designed action space (not related to the performance). The paper proposes LaNAS which is based on an MCTS algorithm to partition the search space into tree nodes by the performance in the tree structure. The performance of the method is shown in the NASBench-101 dataset and Cifar-10 open domain search.

I lean to reject this paper because (1) the motivation is not well justified by the experiments, (2) the comparison on NASBench-101 is not convincing, (3) some important explanation of the method is missing.

Main arguments
The main contribution of the paper is using a learned action space in MCTS rather than a manually designed MCTS algorithm for NAS. However, as far as I know, the MCTS approach for the NAS problem is not a standard solution for NAS (which is not proved to be practically useful in other people’s papers) which diminishes the contribution of the improvements of MCTS in NAS.

Lack of the main comparison. For the motivation of the proposed method, the authors mention the drawbacks of other NAS methods used fixed action space in their RL or MCTS module. However, the authors only show that using a learned action space in MCTS is better than a fixed MCTS algorithm in the experiments. What about using a learned action space in the RL module such as PPO in NAS comparing to the fixed one?

The comparison of NASBench-101 is not convincing. The authors compared with the BO method and claimed that the method is 16.5x more efficient than BO. However, the recently released paper “BANANAS: Bayesian Optimization with Neural Networks for Neural Architecture Search” said that their method is 3.8x more efficient than the one proposed here, which is quite confusing.

Some important explanation of the methods is missing. Throughout the paper, the method to sample from a leaf node is only mentioned in 3.3. However, the corresponding sampling method is unclear. The paper only mentions that MCMC has been adopted to sample from the target subspace. In my opinion, it is not so trivial to use MCMC here and should be elaborated in more detail. Otherwise, people cannot use it.

As given in figure 3, if c is set to be a very small number, the search is similar to simply using a series of predictors and always samples the models with better-predicted accuracies. Is MCTS really useful here? To show the effectiveness of MCTS, it is recommended to experiment on different values of c.

Results given in the upper row of Fig.5 is not useful. In practice, it is already painful to sample about 1000 models and train them for the Cifar-10 dataset. It is more useful to see how this method behaves in the range of (0,1000). However, in Fig.5, different methods are all overlapping in this range and hard to tell whether this method is better than other methods


**Experience Assessment:**

I have published one or two papers in this area.

**Review Assessment: Checking Correctness Of Derivations And Theory:**

I carefully checked the derivations and theory.

**Review Assessment: Checking Correctness Of Experiments:**

I carefully checked the experiments.

**Review Assessment: Thoroughness In Paper Reading:**

I read the paper thoroughly.

---

> ### Author Response · Authors · 2019-11-13
> **thank you, here is our answers**
>
> ==>1. Lack of the main comparison. Only compared manual action MCTS v.s. Latent action MCTS. Want to compare in RL setting.
>
> The focus of this paper is to propose a new MCTS based black-box model for NAS, and we will clarify that in the introduction. Learning actions for RL setting is a nontrivial open problem that deserves separate research work. It is beyond the topic of this paper.
>
> ==>2. The comparison of NASBench-101 is not convincing. Conflict results to “BANANAS: Bayesian Optimization with Neural Networks for Neural Architecture Search” said that their method is 3.8x more efficient than the one proposed here.
>
> Please also refer to “The key limitations of Bayesian Optimization to NAS” in the thread of general answer to all reviewers for more details.
>
> Note that BANANAS compared to AlphaX [1], which is a different work from ours. In the updated version, we have included TPE [2] from Hyperopt, SMAC-random forest [3], and the latest Dragonfly [4] for comparisons. In Fig.5(b), LaNAS substantially outperforms TPE and SMAC up to 12x. Note that we skip Dragonfly in Fig.5 as it is too slow to collect 10^4 samples, instead we compared the performance of Dragonfly, TPE and SMAC in 1000 samples in Fig.6(a)(b). It shows that TPE and BO are better than LaNAS at the beginning (< 500 samples), but LaNAS quickly leads the performance afterwards. This highlights the different focus of algorithms, that LaNAS gives a better result with a bit more budget than TPE and BO.
>
> [1] Wang, Linnan, et al. "Alphax: exploring neural architectures with deep neural networks and monte carlo tree search." arXiv preprint arXiv:1805.07440 (2018).
>
> [2] Bergstra, James S., et al. "Algorithms for hyper-parameter optimization." Advances in neural information processing systems. 2011.
>
> [3] Hutter, F., H. H. and Leyton-Brown, K. “Sequential Model-Based Optimization for General Algorithm Configuration” In: Proceedings of the conference on Learning and Intelligent OptimizatioN (LION 5)
>
> [4] Kandasamy,. "Tuning Hyperparameters without Grad Students: Scalable and Robust Bayesian Optimisation with Dragonfly." arXiv preprint arXiv:1903.06694 (2019).
>
> ==>3. Some important explanation of the methods is missing. The paper only mentions that MCMC has been adopted to sample from the target subspace. In my opinion, it is not so trivial to use MCMC here and should be elaborated in more detail. Otherwise, people cannot use it.
>
> Thank you for your suggestions. We have clarified the sampling part, and provided a few reference implementations to illustrate the concept of “sampling from a polytope”. Since “sampling from polytope” is a well studied ML topic and there are efficient methods to do that, we skip its detailed introduction in this paper and instead provide references and implementations for the clarification.
>
> ==>4. Is MCTS really useful here?
> We have added a new regression tree (RT) model in Fig.5 on 3 datasets, RT is the case that LaNAS removes MCTS and always sample from the most promising node. LaNAS is on average 3x faster than RT, demonstrating MCTS as a valuable component to LaNAS. This is because MCTS adaptively samples in the hyper-space and iteratively builds better models.
>
> ==>5.To show the effectiveness of MCTS, it is recommended to experiment on different values of c.
>
> Great suggestion. In the revision, we have added a new ablation studies on Cp in Fig.7(c), and have achieved a better performance after tuning Cp from 0.5 to 0.1 (updated in Fig.5).
>
> This ablation study shows that Cp is an important factor to the overall performance on NASBench. We also provided an empirical suggestion of Cp = max accuracy of the task * 0.1 based on the ablation studies on 3 NAS datasets. Interestingly, LaNAS is still the leading algorithm on NASBench even at a suboptimal Cp, e.g. 0 or 1.
>
> ==>6. Results given in the upper row of Fig.5 is not useful. It is more useful to see how this method behaves in the range of (0,1000). However, in Fig.5, different methods are all overlapping in this range and hard to tell whether this method is better than other methods
>
> This is an excellent suggestion! We have a better hyper-parameter that can work for [0, 1000] samples than the original scale of samples (Fig.9 in Appendix. A.5), and we have updated the data in Fig.5.
>
> Here is what we have done: we reduce #select from 100 to 10 (to update the latent space more frequently), Cp from 0.5 to 0,1 (according to Fig.7c), #init from 400 to 10 (according to Fig.7(d) that suggests reducing #init improves the performance in [0, 1000]). The new Fig.5 suggests that compared to other approaches, LaNAS starts to lead the performance after 500 samples on NASBench, after 250 samples on ConvNet, and 100 samples on LSTM.
>
> We also added comparisons to BO methods (TPE, SMAC, Dragonfly) in [0, 1000] samples in Fig.6(a),(b). The performance are consistent with Fig.5, that BO and TPE lead the performance < 500 samples, and LaNAS quickly catch up and lead the performance afterwards.

---

### Official Review · AnonReviewer4 · 2019-11-04
**Official Blind Review #4**

**Rating:** 6

**Review:**

The paper describes a new neural architecture search method based on monte-carlo tree search that dynamically adapts the action space.
The methods consists of two stages. In the first stage, the learning stage, the action space is divided into good and bad regions based on a tree structure. In the seconds stage, new data is generated by sampling new architecture with monte-carlo tree search. Those two stages are iterated with new incoming data.

The paper proposes an interesting approach which achieves competitive results compared to state-of-the-art methods.
In general the paper is well written and easy to follow. However,  I haven't fully understood  how the model space is divided at different nodes. What exactly is the splitting criterium? Are the splits axis aligned?

Further comments:

- Figure 4 a and b seemed to be flipped?

- Figure 5 top row would be easier to parse if the x-axis is on a log scale.

- Could you also include other Bayesian optimization methods, such as SMAC or TPE, which should competitive performance on NASBench101 and do not suffer from a cubic scaling



post rebuttal
------------------


I thank the authors for performing additional experiments and clarifying my questions. First of all, I would like to stress that I still think the approach seems promising.
However, I am not entirely convinced by the empirical results.  While the proposed method converges faster to the global optimum than other methods, it is only able to improve by a little epsilon in terms of validation accuracy,  which could indicate overfitting on these tabular benchmarks. Furthermore, in the beginning Bayesian optimisation based methods consistently perform better which might be in practice more relevant for these highly expensive optimisation problems.  Because of that, I keep my initial score.

**Experience Assessment:**

I have published one or two papers in this area.

**Review Assessment: Checking Correctness Of Derivations And Theory:**

I did not assess the derivations or theory.

**Review Assessment: Checking Correctness Of Experiments:**

I assessed the sensibility of the experiments.

**Review Assessment: Thoroughness In Paper Reading:**

I made a quick assessment of this paper.

---

> ### Author Response · Authors · 2019-11-13
> **thank you, here is our answers**
>
> ==>1. I haven't fully understood  how the model space is divided at different nodes. What exactly is the splitting criterion? Are the splits axis aligned?
>
> ---Node level model structures---
> Each node has a linear classifier, W_s*a_i + b_s = acc_i, where W_s, b_s are weights and bias of node s, and a_i is the architecture in model space Ω, acc_i is the accuracy of model a_i after training. Let’s also denote the current collected samples and their accuracies with D_A at node A. We fit the linear classifier with D_A to learn parameters of W_s and b_s.
>
> ---Splitting criterion---
> Since the mean accuracy of currently collected samples at node A is an unbiased estimate of the mean accuracy of search space at Node A (denoted as avg_acc(D_A) ), we used the average accuracy of D_A in splitting the search space. The procedures are:
>
> For any new architecture a_i, if W_A * a_i + b_A > avg_acc(D_A), a_i goes left, and right otherwise.
>
> ---Splitting the hyper-space---
> The aforementioned splitting process requires explicitly calculating directions for a new architecture a_i, and it is not possible for a search space with over 10^17 architectures. Here is the “sampling from polytope” that comes to help.
>
> We illustrate the concept with a concrete example. Essentially we need any architecture a_i satisfy W_A*a_i > avg_acc(D_A) - b_A (denoted as constant C_A) , i.e. W_A*a_i > C_A. Both W_A and C_A are constant after training, and this forms linear constraints for sampling. Given a search path, e.g. A->B->D->H, any architecture a_i must satisfy the 4 constraints of [ W_A*a_i > C_A, W_B*a_i > C_B, W_D*a_i > C_D, W_H*a_i > C_H ]. Conceptually, these constraints in together with predefined constraints on a_i, e.g. 0 <= depth <= 5, 2 <= kernels <= 10, enclose a polytope in hyper-space, and each constraint can be viewed as bifurcating the search space. To sample architectures that satisfy the constraints rendered by a search path, Gibbson sampling or hit-and-run sampling can be used. Please note sampling from a polytope is a well-studied topic in ML, and [1],[2] are two reference implementations.
>
> [1]https://github.com/fontclos/hitandrun
> [2]https://healthyalgorithms.com/2008/11/05/mcmc-in-python-pymc-to-sample-uniformly-from-a-convex-body/
>
>
> ==>2. Could you also include other Bayesian optimization methods, such as SMAC or TPE, which should competitive performance on NASBench101 and do not suffer from a cubic scaling
>
> Excellent suggestions! The updated paper includes comparisons to TPE and SMAC on ConvNet-60K (Fig.5b), and show that LaNAS is still 12.3x, 16.5x more sample efficient than TPE and SMAC-random forest, respectively. Please note we only compared on ConvNet-60K as it has all the accuracies for the architectures in the pre-defined search space. NASBench-420K and LSTM-10K impose constraint of edges <=9 and edges = 9, which renders large number of networks untrained in the search space. Since it is non-trivial to modify HyperOpt (TPE) and SMAC to work with these constraints, it will be unfair to benchmark their performance for sparse rewards. However, we do curate a 6 nodes NASBench to alleviate this problem and compare their performance in the first few hundreds samples in Fig.6(a)(b). Though TPE is faster in the beginning (samples < 500), LaNAS quickly catch up, then lead the performance afterwards as shown by Fig.5(b), Fig.6(a), Fig.6(b).
>
> Please also refer to our answer to “The key limitations of Bayesian Optimization to NAS” in the thread of General answer to all reviewers for more details.

---

### Author Response · Authors · 2019-11-13
**General answer to all reviewers**

Dear reviewers,

We highly appreciate your constructive feedbacks, and we have substantially updated our paper according to your suggestions. All the revisions are highlighted by red.

Here is a list of major revisions in this loop,
1. Adding the context of SMAC, framing the paper under this context. (and still working on)
2. Compared SMAC-random forest, TPE from HyperOpt, and Dragonfly on ConvNet-60k and NasBench-node 6 in Fig.5(b) and Fig.6(a, b).
3. Added Fig.6(b)(c) to demonstrate that optimizing the acquisition function is a key limitation to BO methods.
4. Added Regression Tree (RT) which corresponds to LaNAS without MCTS, and did an ablation study on Cp. Both studies suggest MCTS is a valuable component to LaNAS.
5. Change Fig.5 to log scale, and update the performance data with the newly tuned Cp.
6. Explain how we sample a specific architecture using polytope sampling.

==================Common Questions==================

The key limitations of Bayesian Optimization to NAS

Apart from cubic scaling, BO also has an important limitation in optimizing the acquisition function. As a typical high dimensional search space contains over 10^17 architectures [5], it is not possible to evaluate the acquisition function for every architecture in the search space. However, the current BO NAS methods, e.g. BANANAS (Alg.1 line 3 and line 4), and a peer ICLR submission, “Multi-objective Neural Architecture Search via Predictive Network Performance Optimization” [2], minimize/maximize the acquisition function by iterating through every architecture on the NASBench dataset, which may not be practical in the real setting. In fact, existing SMAC based BO methods, e.g. TPE from hyperopt or Dragonfly, still optimize the acquisition function with an evolutionary algorithm.

To demonstrate the importance of optimizing acquisition to the search performance, we tried to optimize acquisition with TPE and Regularized Evolution under different search budgets for a BO on ConvNet-60K in Fig.6(c)(d). The search budget is the max number of queries to a surrogate model in optimizing the acquisition. We denote the budgets in Fig.6(c)(d) as the percentages of ConvNet dataset. For example, 100% indicates evaluating all the architecture in the search space. As we show in Fig.6(c)(d), the search performance degrades as we reduce the search budgets from 100% to 0.1% for both TPE and RE. Please note 0.1% is still equivalent of 10^14 architectures in a real search space, which is too expensive to enumerate with exhaustive search. Besides, the similar behavior is also observable on NASBench and LSTM. For example, in [2]'s response, they have also seen a significant performance deterioration after reducing the sampling budget from 100% to 0.1% in optimizing the acquisition function.
---------------------------------
ratio  NASBench   LSTM
1           1465.4        558.8
0.1        1564.6      1483.2
0.01      2078.8      1952.4
0.001    4004.4      2984.0
----------------------------------
Here we only tested on NAS datasets, the performance deterioration should be more obvious on a real setting with 10^17 architectures. Therefore, it is unfair to compare BANANAS[1] and [2] to LaNAS without addressing the issue of optimizing acquisition function in a large-scale, high dimensional search space. For SMAC methods, e.g. TPE[3] or Dragonfly[4], that use evolutionary algorithm to optimize the acquisition, LaNAS consistently outperforms them in Fig.5(b), Fig.6(a)(b) on both ConvNet-60K and NASBench-Node6. This is mainly due to the fact that the LaNAS adaptively partitions the search space for sampling.

While BO only uses a fixed kernel which makes it harder to adapt to function with uneven smoothness, LaNAS uses MCTS and online action space learning to dynamically adapt itself towards the important region of the space, and thus achieves better performance in the presence of a decent amount of samples.

[1] White, Colin, Willie Neiswanger, and Yash Savani. "BANANAS: Bayesian Optimization with Neural Architectures for Neural Architecture Search." arXiv preprint arXiv:1910.11858 (2019).

[2] https://openreview.net/forum?id=rJgffkSFPS

[3] Bergstra, James S., et al. "Algorithms for hyper-parameter optimization." Advances in neural information processing systems. 2011.

[4] Kandasamy, Kirthevasan, et al. "Tuning Hyperparameters without Grad Students: Scalable and Robust Bayesian Optimisation with Dragonfly." arXiv preprint arXiv:1903.06694 (2019).

[5]Radosavovic, Ilija, et al. "On Network Design Spaces for Visual Recognition." arXiv preprint arXiv:1905.13214 (2019).

---

### Decision · Program_Chairs · 2019-12-19

**Decision:**

Reject

**Comment:**

This paper proposes an MCTS method for neural architecture search (NAS). Evaluations on NAS-Bench-101 and other datasets are promising. Unfortunately, no code is provided, which is very important in NAS to overcome the reproducibility crisis.

Discussion:
The authors were able to answer several questions of the reviewers. I also do not share the concern of AnonReviewer2 that MCTS hasn't been used for NAS before; in contrast, this appears to be a point in favor of the paper's novelty. However, the authors' reply concerning Bayesian optimization and the optimization of its acquisition function is strange: using the ConvNet-60K dataset with 1364 networks, it does not appear to make sense to use only 1% or even only 0.01% of the dataset size as a budget for optimizing the acquisition function. The reviewers stuck to their rating of 6,3,3.

Overall, I therefore recommend rejection.